# Privacy-Preserving Large Language Model Inference via GPU-Accelerated Fully Homomorphic Encryption

**Leo de Castro**[2,3,*]   **Daniel Escudero**[1,2]   **Antigoni Polychroniadou**[1,2]

[1]J.P. Morgan AI Research   [2]J.P. Morgan AlgoCRYPT Center of Excellence
[3] J.P. Morgan Chase Cybersecurity & Technology Controls
{leo.decastro, daniel.escudero, antigoni.polychroniadou}@jpmorgan.com

## Abstract

As large language models (LLMs) become more ubiquitous, security concerns regarding sensitive queries grow. Due to the complexities of deploying these models, LLM evaluation is often outsourced to a third-party cloud, which leaks the clients' queries to this external provider. These queries could contain sensitive information such as intellectual property, medical information, and proprietary data. Protecting this data while maintaining the LLM's functionality is a major privacy challenge. Fully homomorphic encryption (FHE) presents a natural solution to this problem: simply encrypt the query and evaluate the LLM homomorphically on the cloud machine. The result remains encrypted and can only be learned by the client who holds the secret key. There are two barriers to this solution: (1) FHE operations do not easily support the LLM activation functions and (2) FHE implementations remain too slow to evaluate an LLM in a reasonable time.

In this work, we address both of these barriers to present a fully encrypted version of GPT-2 with forward pass times over $150\times$ faster than the CPU baseline. This result builds on two main technical contributions. First, we present the first open-sourced implementation of GPU-accelerated FHE as an extension to the popular OpenFHE library, achieving roughly $200\times$ performance improvement for many critical functions including bootstrapping. Second, we present novel and extensive experimental analysis of approximations of LLM activation functions to maintain accuracy while achieving this performance. We run extensive benchmarks using the HellaSwag, LAMBADA and ARC datasets, and our results show that the accuracy/perplexity degradation with respect to "out-of-the-box" GPT-2 is minimal.

## 1   Introduction

Large language models (LLMs) have proven to be groundbreaking artificial intelligence tools that are set to change the way humans interact with software. By training on massive amounts of data and using an incredibly large amount of trainable parameters, LLMs are able to provide unprecedented inference results. The tasks that LLMs excel at include natural language generation, question-answering, summarization, translation, code generation, among several others. Models like GPT-3 (https://openai.com/index/gpt-3-apps/) or Claude (https://www.anthropic.com/news/claude-2) can produce coherent and contextually appropriate text on a wide range of topics. However, these models require massive amounts of resources to be trained, and are often not publicly available as this constitutes the provider's intellectual property. This leads to a "inference-as-a-service" scenario, where clients send their queries to external providers who locally run an LLM to return a result to the client. Furthermore, even open source LLMs such as Llama 2

---

[*]This work was done while the author was affiliated with the Massachusetts Institute of Technology.

38th Conference on Neural Information Processing Systems (NeurIPS 2024).

(https://llama.meta.com/llama2/) are very expensive to run in commodity hardware and still require in most cases delegating inference to a third party provider.

Unfortunately, delegating inference is undesirable in many settings where the client wants to preserve the privacy of their input. Furthermore, as mentioned above, there are multiple contexts in which the model owner also wants to retain privacy of the model itself, for example when the model involves massive monetary resources to be trained, or when it incorporates sensitive data (*e.g.* a bank servicing a credit score model trained on internal data). This is particularly relevant as LLMs become more pervasive and find more use-cases that permeate all areas of society. This tension between privacy and utility heavily limits the applicability of LLMs, rendering them useless in contexts where data cannot be outsourced due to privacy constraints.

Towards resolving this tension, fully homomorphic encryption (FHE) is a promising tool that enables computing on data without revealing it, only outputting the final result (*cf* [Mar+22] for a survey). Using FHE, a client can *encrypt* their query to the server, who can locally apply their model to this encrypted data, making use of the homomorphic properties of the scheme to obtain an encrypted result, which is sent back to the client for decryption. See Fig. 1a for a pictorial representation of this interaction pattern. Advances in the last decade on all fronts including algorithms, software and hardware, have made FHE practical for several tasks that were not within reach before. However, LLMs are in an entirely different regime: their computation is already very expensive in the clear, up to the point in which specialized software such as high-end GPUs, coupled with several architectural optimizations, are needed in order to provide a reasonable inference latency. Any computation that is ran under FHE becomes *much* slower, which is going to be a major blocker when porting LLMs to FHE. However, the question remains:

*How practical is FHE-based privacy-preserving LLM evaluation?*

To address this question, a good starting point is the CKKS scheme by Cheon et al. [Che+17], which enables approximate additions and multiplications over real (in fact, complex) numbers. We provide detail background on FHE and CKKS in Sections 2.2 and C.1, respectively. The literature in improving the efficiency of this scheme is vast and fruitful [Bos+21; HK20; Jun+21], and this has enabled several applications in contexts such as logistic regression [Che+18a] and secure password search [Che+18b].

Only the recent work of Zhang et al. [Zha+24] has explored large language model inference via CKKS, reporting an implementation of the transformer architecture in C++, using the SEAL library for FHE (https://github.com/Microsoft/SEAL). Their experiments report minor accuracy degradation due to polynomial approximations needed in FHE, and performance in Intel CPUs seems promising, as it is accelerated via HEXL [Boe+21]. We discuss this work further in section 1.2. Although promising given the massive overheads involved in both LLMs and FHE, this is still far from practical for real-world usage, even for applications that are not latency sensitive such as text summarization or content generation (in contrast to chatbots or Q/A tasks, which are more demanding in terms of responsiveness).

## 1.1 Our contributions

We approach the problem of improving the efficiency of FHE-based privacy-preserving LLM inference, by providing a *GPU-based* implementation of the transformer architecture using CKKS. Prior work [Jun+21] has shown GPUs to help in improving the efficiency of CKKS. However, to the best of our knowledge, there is currently no available implementation of such works to deploy and test these ideas. In contrast, there are popular open-source *CPU-based* frameworks that aim at making FHE techniques more accessible by providing high level programming interfaces, and access to multiple FHE schemes, like CKKS. One such framework is *OpenFHE* [AB+22], which has gained traction as one of the most comprehensive and widely used FHE implementations available. Unfortunately, OpenFHE is limited to CPUs, and hence its performance in tasks such as LLM inference would be rather poor.

In this work we extend the capabilities of OpenFHE by enabling a GPU-based workflow, which leads to direct efficiency improvements across many FHE applications that build on this framework—not only LLMs. This requires a deep understanding of the internal CKKS operations to replicate them on

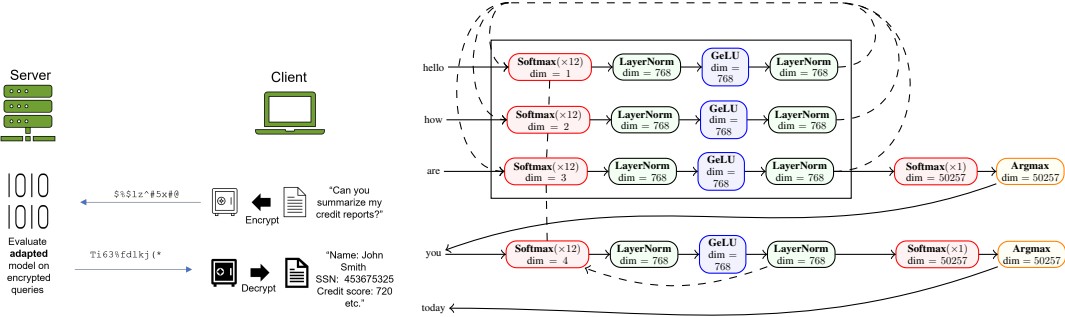

(a) Interaction pattern in outsourced privacy-preserving LLM evaluation. Here, a client (right side) encrypts (say) financial reports to the server (left side), who receives an unintelligible ciphertext. The server evaluates the *adapted* model—incorporating approximations and working with ciphertexts instead of cleartext values—and sends back an encryption to the client. Finally, the client, who is the only party who knows the secret-key, can decrypt this message and learn the result.

(b) Overview of the GPT-2 inference flow. The diagram only shows the blocks that are expensive to run in FHE, ignoring simpler operations such as linear or affine layers. The input is the sentence "hello how are", and the completion is "you". To get the next token, only the path associated to the lastest token "you" needs to be computed, which in this example leads to the token "today". The vertical lines between the leftmost softmax boxes illustrates that each new softmax is somehow dependent on the inputs of previous ones. Every block is labeled with the dimension of the input it takes. For softmax, the number in parentheses represents how many such calls are made.

Figure 1: On the left: communication pattern between the two parties. On the right: GPT-2 architecture, which corresponds to the local computation by the server.

the GPU. We provide benchmarks for our GPU-accelerated CKKS bootstrapping in appendix E. We have open-sourced the code of our OpenFHE+GPU extension[2].

With our GPU-based implementation in place, we set out to benchmark the performance of large language models under FHE. We focus specifically on the GPT-2 architecture by OpenAI, which is fully open-source and shares common features with many of the more powerful industry-grade models. One first obstacle we face is that FHE techniques do not support all operations available to a common CPU/GPU and instead only supports additions and multiplications. As usual in the FHE literature, we use off-the-shelf polynomial approximations to replicate as faithfully as possible the transformer architecture, while adapting for FHE use. Our approximations are discussed in Appendix D. Note that these modifications have the potential of negatively affecting the accuracy of the model, which is far from ideal. To address this, we modify the GPT-2 implementation from HuggingFace's transformers library (https://github.com/huggingface/transformers) so that it includes these FHE-friendly modifications, and thoroughly benchmark the resulting accuracy using the LM evaluation harness library (https://github.com/EleutherAI/lm-evaluation-harness) on a selection of tasks. This allows us to select optimal parameters for the approximations that strike the right balance between efficient FHE runtimes and model accuracy. Furthermore, for reproducibility we also open source our modified HuggingFace GPT-2 implementation.

Our results given in section 3 show that a GPU-accelerated FHE implementation provides a roughly $200\times$ speedup in the GPT-2 forward pass, reducing the time from several hours to just a few minutes. This brings the forward pass time down to a range where non-real-time applications become more practical, such as document summarization and fine-tuning models on private data.

## 1.2 Related Work

There is a long line of works studying secure inference for protecting the privacy of both a client owning a query, and a server holding a trained model. At a high level, we can divide these techniques into two groups: highly interactive approaches based on MPC, and less communication-demanding but more computationally-heavy paradigms based on FHE. We focus this section on FHE-based approaches, leaving the discussion on MPC-based techniques to Appendix A.

---

[2]https://github.com/leodec/openfhe-gpu-public

**FHE-based LLM inference.** FHE-based secure inference has the notable advantage that it preserves the same communication pattern of non-private inference: the client sends the query to the server, who performs certain (presumably heavy) computation and sends back the result. This is applicable to real-world settings where client and servers may not be well connected, and the server is considerably more powerful than the client. In this context, the most relevant work in secure LLM inference with FHE is [Zha+24]. This work makes use of several polynomial approximations from the literature, some of which we borrow as well (see Appendix D). Importantly, their implementation is limited to CPU, which caps their performance substantially. Rather than comparing to this work, we instead compare directly to the out-of-the-box OpenFHE CPU implementations of the FHE functions. This allows us to account for variations in the approximations and the placement of the bootstrapping functions.

The work of [Zim+23] studies HE-friendly approximation of the transformer architecture, but it is not applicable to our case since this require re-training. Primer [ZLJ23] and THE-X [Che+22] also employ FHE for LLM evaluation (Primer in fact mixes FHE and MPC), but these works also make substantial modifications to the underlying model. THE-X even reveals intermediate values of the computation.

**Privacy-preserving ML for other models.** Finally, we mention that there are several other works that have studied FHE-based inference of other machine learning models, such as convolutional neural networks (*cf.* [AB24; Boe+19; GB+16; JVC18]). These are not applicable to transformers directly as they do not support all of the operations involved in this architecture, and additionally the scale of the models they consider is much more reduced.

## 1.3 Setting and Threat Model

We consider a client who holds as input a text sequence, and a server who holds a large language model. The goal is for the client to learn the evaluation of their query on the model without leaking the input to the server, and while protecting the privacy of the model towards the client. See Fig. 1a for a pictorial representation of the task and the communication flow. The server does not learn any information about the client's input, but we provide no correctness guarantees regarding the result the server returns to the client—a corrupt server can return an incorrect answer, or no answer at all. This is consistent with prior works, and it is strictly better than the guarantees provided by MPC-based solutions, which may leak information towards a corrupt server that deviates from the protocol specification.

We assume the client has access to the *tokenizer* of the model (see Section B.1), so that the client can locally transform their text into a sequence of real-valued vectors, which are then encrypted towards the server. We do not provide any guarantees on the plaintexts underlying the ciphertexts that the client sends. In particular, a corrupt client may send a sequence of vectors that does not correspond to valid token embeddings, and will be able to learn the LLM evaluation on this input. This is in par with previous privacy-preserving ML works based on FHE.

## 2 Preliminaries

In what follows we provide background on large language models and fully homomorphic encryption.

Some general notation we will use throghout the paper is the following. Vectors are denoted by bold letters, like $x$, and indexing the $i$-th entry is denoted by $x[i]$. Given a positive integer $n$, we let $[n]$ denote the set $\{1, \ldots, n\}$.

### 2.1 Large Language Models

A large language model (LLM) is a type of machine learning (ML) model that is characterized by its ability to predict *language*, with the "large" term emphasizing their comparatively gigantic sizes and computational demands. Vaswani et al. [Vas+17] introduced the transformer architecture, which is the basis for several LLMs that came right after. Among LLMs, an interesting and relevant family are generative pretrained transformers (GPTs), which are used in natural language processing contexts. This family, developed by OpenAI, has been widely influential and has spawned a series of follow-ups. In this work we focus specifically on the **GPT-2** model, which is trained on WebText:

40 GB of text, 8 million documents, from 45 million webpages upvoted on Reddit. We chose this model as (1) it is fully *open source*, (2) it follows the transformer architecture shared by other more powerful LLMs, and (3) this is already challenging in terms of efficiency for current FHE approaches. We note however that our findings carry out to several other LLMs that follow this paradigm, such as the larger models like GPT-3 or GPT-4 or other transformer-based LLMs like Llama and Llama 2. In what follows, we describe the GPT-2 architecture in detail. There are four variants of GPT-2 which vary in size and performance: S, M, L and XL, and we discuss below the points where these differ.

LLMs use deep learning to analyze and generate human-like text. The transformer architecture by Vaswani et al. [Vas+17] receives as input a piece of text, which is split into numerical representations referred to as *tokens*. Transformers are comprised of an enconder and a decoder section, which are very similar in structure. However, generative LLMs such as GPT are *decoder-only*, and so for the sake of this work we will focus on the decoder component of the transformer architecture; we note that encoders follow a similar structure and our findings apply to encoder-decoder or encoder-only architectures as well.

The model is trained to predict the best next word given a sequence of words. For example, it may receive as an input "Today is a good", and then predict "day" as the next word. The resulting concatenated sentence "today is a good day" can be fed into the model again to obtain as the next word, perhaps, "for". This way a sequence like "today is a good day for running outside" can be generated.

An overview of the GPT-2 architecture, highlighting the blocks that are most relevant for FHE, is given in fig. 1b; see appendix B for additional details. Throughout this work, we use the "small" variant of GPT-2 with embedding dimension $d = 768$.

## 2.2 Fully Homomorphic Encryption

A fully homomorphic encryption (FHE) scheme [RAD+78], [Gen09] is an encryption scheme that allows computations to be performed over the data while the data remains encrypted. More formally, an FHE scheme is defined by the following tuple of algorithms.

- $(\mathsf{sk}, \mathsf{pk}, \mathsf{evk}) \leftarrow \mathsf{KeyGen}(1^\lambda)$. This is the key generation algorithm. The input is the security parameter $\lambda$ and the output is three keys. The secret key $\mathsf{sk}$ is used for decryption, the public key $\mathsf{pk}$ is used for encryption, and the evaluation key $\mathsf{evk}$ is used to homomorphically compute over encrypted data.
- $\mathsf{ct} \leftarrow \mathsf{Encrypt}(\mathsf{pk}, m)$. This is the encryption algorithm. It takes in a message $m$ and a public key $\mathsf{pk}$ and outputs a ciphertext $\mathsf{ct}$.
- $m' \leftarrow \mathsf{Decrypt}(\mathsf{sk}, \mathsf{ct}')$. This is the decryption algorithm. It takes in a ciphertext $\mathsf{ct}'$ and a secret key $\mathsf{sk}$ and outputs a message $m'$.
- $\mathsf{ct}_f \leftarrow \mathsf{Eval}(\mathsf{evk}, \mathsf{ct}, f)$. This is the homomorphic evaluation algorithm. It takes in as input an evaluation key $\mathsf{evk}$, a ciphertext $\mathsf{ct}$, and a function $f$. Let $m$ be the message encrypted by $\mathsf{ct}$ (i.e. $m \leftarrow \mathsf{Decrypt}(\mathsf{sk}, \mathsf{ct})$). The output of $\mathsf{Eval}$ is the ciphertext $\mathsf{ct}_f$ that encrypts $f(m)$.

FHE must satisfy the same security level as a regular encryption scheme, which dictates that a party without access to the secret key cannot distinguish between encryptions of any two messages, even if the messages are adversarially chosen.

## 3 Experimental Results

In this section, we present the full LLM runtimes under FHE. These evaluations are run entirely on the server, and at no point can the server view the underlying query or any intermediate value. Furthermore, the output of the LLM forward pass can be fed directly back into the model to compute the next token without any interaction with the client. This powerful technique allows an arbitrary number of forward passes to be executed on the client's encrypted query. This method extends to other operations that require the forward-pass as a subroutine, such as fine-tuning on private data.

As we mentioned in the introduction, we focus on GPT-2 (small) due to its accessibility as well as the similarity in the architecture of larger GPT models. Our performance benchmarks can be extended to models with many more parameters by linearly scaling the transformer architecture.

| Benchmark | Lambada | | HellaSwag | ARC (Easy) |
| --- | --- | --- | --- | --- |
| | Perplexity | Accuracy | Accuracy | Accuracy |
| Baseline | 40.0554 | 0.3256 | 0.2892 | 0.4381 |
| Approximate | 41.8580 | 0.3013 | 0.2918 | 0.4327 |

Table 1: Performance of GPT-2 (small) with our different approximations vs. the unaltered baseline. We use polynomials of degree 4—each composed twice—for the comparison approximations, (see Section D.1). We use 16 Newton iterations for the inverse square root (see Section D.3). We use 7 iterations of Goldschmidt algorithm for the Softmax division, and we use $r = 7$ for the approximation of exp in Softmax (see Section D.4).

| Function | SoftMax | LayerNorm | GeLU | Argmax |
| --- | --- | --- | --- | --- |
| depth | 133 | 13 | 17 | 272 |
| number of ciphertexts | 0.25 | 1.5 | 6 | 1 |

Table 2: Depths of our approximate activation functions. The approximations (described in appendix D) have the same parameters as the plaintext circuits benchmarked in table 1. The softmax input size is $128 \times 128$ values, which requires a in a depth-7 comparison tree to compute the max of all sets of 128 values in parallel. The number of slots in each ciphertext is $n = 2^{16}$. Non-integer ciphertexts indicate that not all slots are filled and batched evaluation is available in this layer.

## 3.1 Accuracy of the Approximate Model

In order to make our LLM compatible with FHE, we replace each non-linear function with the corresponding approximation described in section 2. We evaluate this variant of GPT-2 on standard accuracy benchmarks to ensure that these approximations do not compromise the model's performance. We achieve this by forking the GPT-2 implementation in the HuggingFace transformers library (https://github.com/huggingface/transformers), and making the following modifications in order to reflect the changes that FHE imposes:

- The GeLU activation is replaced by the approximation from Section D.2. We use degree 2 for the $f$ and $g$ polynomials in the comparison from Section D.1, and we compose them 2 times each.

- LayerNorm is approximated as in Section D.3. We use 16 Newton iterations

- SoftMax is approximated as in Section D.4. For the approximation of exp we use $r = 7$, and for Goldschmidt algorithm—used for the division—we use 7 iterations.

Performing these modifications is intricate as the transformers library is not intended to support changes such as replacing the SoftMax, for instance, which is rather uncommon in machine learning contexts. Once our modified model is loaded in HuggingFace's "format", we are able to leverage the Language Model Evaluation Harness library (https://github.com/EleutherAI/lm-evaluation-harness), which includes multiple benchmarks to evaluate LLM performance. Our accuracy benchmarks appear in table 1, where we measure the performance of our modifications with respect to the baseline GPT-2 (small) on three datasets: Lambada, HellaSwag and ARC. The Lambada dataset is a collection of passages and sentences used for evaluating the ability of language models to understand context and perform coherent text continuation or next word prediction. HellaSwag tests LLM's ability to capture commonsense reasoning about situations described in natural language. ARC (AI2 Reasoning Challenge) is a dataset created by the Allen Institute for Artificial Intelligence (AI2) to evaluate question answering systems' ability to perform multi-step reasoning. We refer the reader to the evaluation harness library for details on these tasks.

Overall, we observe that our modifications incur in little accuracy degradation with respect to the baseline model. This reflects the robustness of large language model to slight deviations, highly exploited in the quantization literature (cf [Zhu+23]), and is crucial for enabling privacy-preserving inference. Note that these approximations are also useful for MPC-based approaches.

## 3.2 Runtimes of LLM Inference in FHE

We now present the end-to-end runtime of a GPT-2 forward pass using our GPU-accelerated FHE. Note that, as illustrated in Fig. 1b, the complexity of a GPT forward pass is dependent on the position of the token being generated in the output, given that the dimension for the softmax in each decoder block depends on the token position. Furthermore, as we discuss in Remark 2, all tokens of the input sequence have to be processed *once* by the decoder blocks before any new token can be generated. Throughout this section, we benchmark generating a token at position 128, assuming that the previous input tokens have been processed. The cost of processing the input is amortized away as more tokens are produced, which is also consistent with prior works.

We note a few important optimizations that are incorporated into this benchmark:

*Input & Output Sizes.* We give the depth of each approximation in table 2. Recall from the high-level GPT architecture that SoftMax and GeLU are run once per block and LayerNorm is run twice per block. The GPT-2 model consists of 12 blocks, and the final ArgMax function is run at the end of the forward pass. The dimension of one token embedding is 768, and the inputs and outputs of both LayerNorm operations is $128 \times 768$. The GeLU input consists of 24 channels of the typical $128 \times 768$, resulting in a total input of $3072 \times 768$. By contrast, the SoftMax input is the result of many inner-product operations with the context embeddings, resulting in an input and output size of $128 \times 128$. With $2^{16}$ slots in each ciphertext, this gives the values in the second row of Table 2.

*Batched Evaluation.* When a function is evaluated over an input that does not use all available slots in a ciphertext, additional performance can be gained by evaluating another input to that function and using the additional unused slots. This batched evaluation maximizes the available parallelism in the CKKS scheme. For example, the LayerNorm function only requires 1.5 ciphertexts to store the input and output. If only one LayerNorm function is being evaluated, then we must perform the operation over two ciphertexts even though the second is half empty. However, if we have the option of running a second LayerNorm function over an independent input, we can evaluate both LayerNorm functions using only three ciphertexts, which doubles our throughput with only a $50\%$ increase in latency. This is an important optimization for tasks such as training or fine-tuning, where the model is evaluated on batches of samples from the training set. We also present the "unbatched" single-input evaluation for comparison.

**Benchmarks.** We present our benchmarks in Figure 2 and Figure 3. Both figures display the forward pass time of our encrypted GPT-2 at position 128. All individual layer benchmarks include the internal bootstrapping time, which is interleaved within the function as needed. All benchmarks were run on the same machine as the bootstrapping benchmarks in appendix E. This machine has an Intel Xeon chip running at 2.4 GHz and 2 TB of RAM as well as an NVIDIA A100 80GB PCIe.

In Figure 2, we demonstrate the speedup of our GPU-accelerated FHE library when applied to the task of a GPT-2 forward pass. This figure measures our GPU implementation against the out-of-the-box OpenFHE functions running on a CPU.

In the unbatched forward pass, the SoftMax function is one of the most expensive operations primarily due to the low utilization of the ciphertext. When switching to batched evaluation, the overhead of the SoftMax drops significantly ($4\times$) as well as the LayerNorm function discussed above. The GeLU function has full utilization of the ciphertexts, so the overhead with batching remains the same. The batching speedups translate into the benchmarks for the full model. Recall that the full forward pass consists of 12 blocks and an ArgMax. We do not batch the ArgMax evaluation since only a small portion of the ciphertext is left unused.

We provide benchmarks at two different security levels depending on the application requirements. Setting the security parameter $\lambda = 128$ is standard for encryption schemes, although many applications allow a slightly weaker $\lambda = 80$. Concretely, setting $\lambda = 128$ gives us a bootstrapping routine that refreshes 20 ciphertext levels in roughly 550 milliseconds, while relaxing to $\lambda = 80$ allows a bootstrapping routine that refreshes 45 levels in under 1 second. This increase in the bootstrapping throughput is the main source of speedup.

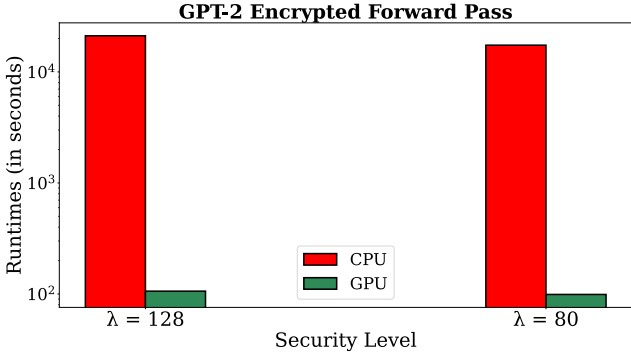

Figure 2: This figure presents benchmarks of our GPT-2 forward pass running under FHE. The polynomial approximations for the activation functions as well as the high-level bootstrapping algorithm are identical in both benchmarks. The CPU bar uses the out-of-the-box OpenFHE functions, while the GPU bar uses our GPU-accelerated implementation. Both benchmarks are for a single (unbatched) evaluation. The speedup when switching to the GPU is about $150\times$.

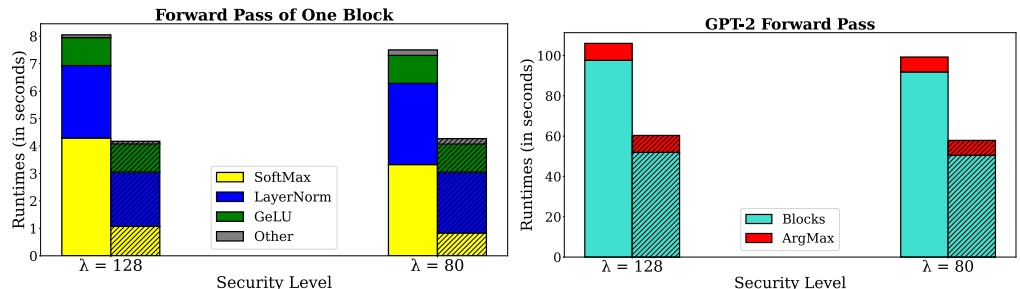

Figure 3: This figure presents the GPU-accelerated encrypted GPT-2 forward pass runtimes for generating a token at position 128. The hatched bars indicate the batched evaluation times, where unused ciphertext slots are filled with independent evaluations (also, see description in paragraph "batched evaluation", p.7). The savings are maximized with four independent evaluations, allowing the SoftMax to be fully utilized. The full forward pass consists of 12 blocks followed by an ArgMax.

### 3.3 Limitations

We briefly discuss the limitations of our results. Our benchmarks are based on the accuracy of the GPT-2 model with the activation functions replaced with polynomial approximations. The degree of these polynomials has a major impact on the performance of the encrypted forward pass, since a higher degree directly translates into deeper circuits that require more bootstrapping operations. While many LLM models seems to remain accurate with low precision, many other AI models such as image recognition models require higher precision during evaluation to maintain accuracy. If a model requires a higher precision than GPT-2, the polynomial approximations would need to be increased. When the required precision increases beyond roughly 16 bits, the complexity of the bootstrapping itself must be increased, since internal to the bootstrapping is an approximation of a modular reduction function. The relatively low precision required by these transformer models is crucial to our results.

## Acknowledgments

We would like to thank Marzyeh Ghassemi for generously providing computational resources for this project.

This paper was prepared in part for information purposes by the AI Research Group, the AlgoCRYPT Center of Excellence, and Cybersecurity & Technology Controls group of JPMorgan Chase & Co and its affiliates (JP Morgan), and is not a product of the Research Department of JP Morgan. JP Morgan makes no representation and warranty whatsoever and disclaims all liability, for the

completeness, accuracy or reliability of the information contained herein. This document is not intended as investment research or investment advice, or a recommendation, offer or solicitation for the purchase or sale of any security, financial instrument, financial product or service, or to be used in any way for evaluating the merits of participating in any transaction, and shall not constitute a solicitation under any jurisdiction or to any person, if such solicitation under such jurisdiction or to such person would be unlawful.

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

# Supplementary Material

## A    Related Works on MPC-based LLM inference

A paradigm that has received more attention in the literature is secure inference based on MPC, where the client and the server interact in a communication-heavy protocol in order to privately compute the prediction. The computation for *both* server and client is proportional to the computation that would happen in cleartext, which is orders of magnitude more lightweight than FHE, but places computational overheads on the client that do not appear with FHE. Furthermore, this benefit in computation is only relevant if communication is not a bottleneck, which only occurs in contexts where the client and the server share a very fast connection. This is not the case for realistic scenarios, which makes of MPC a less useful solution for this use-case.

Iron [Hao+22] is one of the first works in exploring secure transformer inference using secret-sharing-based two-party computation. Sigma [Gup+23] is another two-party protocol based on the *preprocessing model*, where the two parties are assumed to have correlated randomness which is independent of their inputs, for free. In this setting, the authors show that it is possible to evaluate GPT-2 within 2 seconds. However, this does not include the cost of generating the preprocessing (which is by far the main bottleneck for two-party protocols based on secret-sharing), and moreover these results are for a very strong network with 9.4 Gbps bandwidth and 0.05 ms ping. CipherGPT [Hou+23] provides full benchmarks that also include the preprocessing time, and we see from [Hou+23, Table 5] that it takes up to 25 mins to generate on token in their framework—even assuming fast networking conditions with bandwidth 377 MBps and RTT of 0.8ms. MPCFormer [Li+22] uses knowledge distillation to better approximate the transformer architecture using MPC-friendly building blocks, which is a promising direction; however, our focus was on evaluating out-of-the-box architectures such as GPT-2. Bumblebee [Lu+23] improves over Iron, but again it has very strong networking requirements (1 gigabit per second (1 Gbps), and a ping time of 0.5ms). BOLT [Pan+24] also improves over Iron and presents similar communication bottlenecks. Finally, the work of Puma [Don+23] approaches larger models (Llama-7B), but it does so in the *three*-party setting with honest majority, which is substantially simpler and arguably less realistic than two parties.

## B    LLM Architecture Background

### B.1    Tokenization and Embedding.

First, the input sentence is split into the so-called *tokens*, which roughly correspond to words, using a technique known as Byte-Pair Encoding (BPE) [Gag94]. GPT-2 recognizes 50257 different tokens, which is its so-called *vocabulary size*. The *window size* determines the maximum amount of tokens that can be handled, and it is set to $1024$ in GPT-2. Finally, each token is mapped—via some pre-trained mapping—to an *embedding vector* of some dimension $d$ which depends on the model version: 768 for S, 1024 for M, 1280 for L and 1600 for XL. The result is a sequence of vectors $(\boldsymbol{u}_1, \boldsymbol{u}_2, \ldots, \boldsymbol{u}_N)$, for some $N \leq 1024$.

**Remark 1** *As in all previous works on privacy-preserving LLM inference, we assume the tokenizer is known by the client, so that the client can compute on their own the embedding $(\boldsymbol{u}_1, \boldsymbol{u}_2, \ldots, \boldsymbol{u}_N)$. This is the actual client's input to the computation.*

**Positional Embedding.**    Each of the embedding vectors $\boldsymbol{u}_i \in \mathbb{R}^d$ is added to a pre-trained *positional encoding* vector $\boldsymbol{t}_i \in \mathbb{R}^d$ that depends on the position $i$ that the associated token has in the sequence. This is intended to model the different positional relationships between words. The result $\boldsymbol{x}_i = \boldsymbol{u}_i + \boldsymbol{t}_i \in \mathbb{R}^d$ is the "final embedding" of the $i$-th token.

### B.2    Decoder

The input sequence is fed into a decoder stack, which produces as a result what it believes to be the best next token given the current input. This stack is obtained by iterating several decoder blocks, with the number of iterations depending on the GPT-2 model version: 12 for S, 24 for M, 36 for L

and 48 for XL. The decoder block takes as input a sequence $(\boldsymbol{x}_1, \ldots, \boldsymbol{x}_N) \in (\mathbb{R}^d)^N$, and performs the steps detailed below.

**Multi-head masked self-attention.** Each decoder block has a *query matrix* $\boldsymbol{Q} \in \mathbb{R}^{d \times d}$, a *key matrix* $\boldsymbol{K} \in \mathbb{R}^{d \times d}$ and a *value matrix* $\boldsymbol{V} \in \mathbb{R}^{d \times d}$. For each embedding $\boldsymbol{x}_i$ with $i \in [N]$, we derive the following:

- A *query vector* $\boldsymbol{q}_i = \boldsymbol{x}_i^\mathsf{T} \boldsymbol{Q} \in \mathbb{R}^d$; this is split into $d/64$ vectors $\boldsymbol{q}_i^{(1)}, \ldots, \boldsymbol{q}_i^{(d/64)} \in \mathbb{R}^{64}$
- A *key vector* $\boldsymbol{k}_i = \boldsymbol{x}_i^\mathsf{T} \boldsymbol{K} \in \mathbb{R}^d$; this is split into $d/64$ vectors $\boldsymbol{k}_i^{(1)}, \ldots, \boldsymbol{k}_i^{(d/64)} \in \mathbb{R}^{64}$.
- A *value vector* $\boldsymbol{v}_i = \boldsymbol{x}_i^\mathsf{T} \boldsymbol{V} \in \mathbb{R}^d$. this is split into $d/64$ vectors $\boldsymbol{v}_i^{(1)}, \ldots, \boldsymbol{v}_i^{(d/64)} \in \mathbb{R}^{64}$.

Now, for every $\ell \in [d/64]$, the $i$-th embedding is scored against the $j$-th embedding for every $j \le i$.[3] This is done by taking the dot product $\langle \boldsymbol{q}_i^{(\ell)}, \boldsymbol{k}_j^{(\ell)} \rangle$, turning these into weights via softmax (and normalizing by $\sqrt{64} = 8$) as

$$\forall i \in [N], \forall \ell \in [d/64]: \quad (\lambda_{i1}^{(\ell)}, \ldots, \lambda_{ii}^{(\ell)}) \leftarrow \mathsf{SoftMax}(\underbrace{\langle \boldsymbol{q}_i^{(\ell)}, \boldsymbol{k}_1^{(\ell)} \rangle / 8, \ldots, \langle \boldsymbol{q}_i^{(\ell)}, \boldsymbol{k}_i^{(\ell)} \rangle / 8}_{\text{length } i}).$$

The resulting "weighted value" associated to the $i$-th token is then $\boldsymbol{z}_i^{(\ell)} = \sum_{j \le i} \lambda_{ij}^{(\ell)} \cdot \boldsymbol{v}_j^{(\ell)} \in \mathbb{R}^{64}$. The final output for the $i$-th token of the multihead self-attention section is the projected concatenated vector $\boldsymbol{z}_i := (\boldsymbol{z}_i^{(1)} \| \cdots \| \boldsymbol{z}_i^{(d/64)}) \cdot \boldsymbol{M} \in \mathbb{R}^d$, where $\boldsymbol{M} \in \mathbb{R}^{d \times d}$ is a pre-trained projection matrix.

**Residuals and layer normalization.** For $i \in [N]$, the output $\boldsymbol{z}_i \in \mathbb{R}^d$ is added to the original input $\boldsymbol{x}_i \in \mathbb{R}^d$ to the decoder, and then the result $\boldsymbol{w}_i = \boldsymbol{z}_i + \boldsymbol{x}_i$ is fed into a *layer normalization* step which consists of

$$\mathsf{LayerNorm}(\boldsymbol{w}_i) := \gamma \cdot \frac{\boldsymbol{w}_i - \mu}{\sqrt{\sigma^2 + \epsilon}} + \beta.$$

Here $\gamma, \beta \in \mathbb{R}$ are learnable parameters, and $\mu$ and $\sigma^2$ are the mean and variance, which are defined as $\mu = \frac{1}{d} \cdot \sum_{j=1}^d \boldsymbol{w}_i[j]$ and $\sigma^2 = \frac{1}{d} \cdot \sum_{j=1}^d (\boldsymbol{w}_i[j] - \mu)^2$. The value $\epsilon$ is a fixed small constant to avoid division by zero. Let us denote the resulting vector $\mathsf{LayerNorm}(\boldsymbol{w}_i)$ by $\boldsymbol{z}_i' \in \mathbb{R}^d$.

**Feed-forward Neural Network and Decoder Block Output.** The normalized vector $\boldsymbol{z}_i' \in \mathbb{R}^d$ from before is passed to a neural network made of two layers. The first is a product with a $d \times 4d$, followed by GeLU, and the second layer projects back to the initial dimension by multiplying with a $4d \times d$ matrix. The result $\boldsymbol{y}_i \in \mathbb{R}^d$ is then added to $\boldsymbol{z}_i'$, and the final output of the decoder block is taken to be $(\boldsymbol{x}_1', \ldots, \boldsymbol{x}_N') \in (\mathbb{R}^d)^N$, where $\boldsymbol{x}_i' = \mathsf{LayerNorm}(\boldsymbol{z}_i' + \boldsymbol{y}_i)$. This is the sequence that gets passed through a new decoder block, iterating the process a number of times dependent on the GPT-2 version.

### B.3 Final layer

Suppose that $(\boldsymbol{t}_1, \ldots, \boldsymbol{t}_N) \in (\mathbb{R}^d)^N$ is the output of the final decoder. The vector $\boldsymbol{t}_N \in \mathbb{R}^d$ is fed into a linear layer $\boldsymbol{t}_N \cdot \boldsymbol{L}$, where $\boldsymbol{L} \in \mathbb{R}^{d \times \texttt{vocab\_size}}$ is a pre-trained matrix.[4] Followed by this there is a $\mathsf{Softmax}$ layer, which outputs a vector of logits

$$\boldsymbol{b}_N \leftarrow \mathsf{Softmax}(\underbrace{\boldsymbol{w}_N \cdot \boldsymbol{L}}_{\text{length } \texttt{vocab\_size}}) \in \mathbb{R}^{\texttt{vocab\_size}}.$$

Here, `vocab_size` is the vocabulary size, which as we have mentioned equals 50257 for the case of GPT-2.

---

[3]This is where the "masked" name comes from: in standard self-attention—used by the encoders in the transformer architecture—a given token is scored against *all* other tokens. In the decoding self-attention each token is only scored against the *previous* tokens to it, effectively "masking" the future ones.

[4]The vectors $\boldsymbol{t}_1, \ldots, \boldsymbol{t}_{N-1}$ for the *final* decoder block iteration are not needed and do not need to be computed in a first place.

**Multiple decoding methods.** The output $b_N \in \mathbb{R}^{\texttt{vocab\_size}}$ is interpreted as a mapping $[\texttt{vocab\_size}] \to [0,1]$ indicating, for each possible token index, how likely it is for this token to be the next in the sequence. There are multiple decoding methods to map this vector to an actual token. The simplest one—which is the approach we take in our work—consists of selecting $i^* \in [\texttt{vocab\_size}]$ as the $\mathsf{ArgMax}$ of $b_N$. Other approaches such as beam search or top-k sampling exist, but their implementation in FHE becomes substantially more complex (with the $\mathsf{ArgMax}$ already presenting noticeable challenges; see Section 3 for details).

**Remark 2 (On computing subsequent tokens)** *Once the next token index $i^*$ has been determined, the embedding vector corresponding to index $i^*$ can be fetched from the same pre-trained index-to-embedding table used initially when mapping the client's input. Then, this vector can be used as the $N+1$'th embedding $x_{N+1}$, and the updated sequence $(x_1, \ldots, x_N, x_{N+1})$ can be processed in the same way as before to produce the next token.*

*Crucially, note that many of the intermediate values produced during the generation of the $N+1$-th token can be* reused *for producing the $N+2$-th token. For the self-attention layers, the iteration $i \in [N+1]$ only needs to be done for $i = N+1$, since for $i \in [N]$ all the values involved only depend on tokens in positions $j$ with $j \leq i$, which have already been processed. In particular, only $d/64$ SoftMax calls are needed, instead of $N \times d/64$. This is crucial for our FHE solution, since this means that the cost of generating the* first *(additional) token is not the same as for subsequent ones, which are about a factor of $N$ cheaper to compute.*

## C  CKKS Background

### C.1  The CKKS FHE Scheme

In this work, we use the CKKS FHE scheme [Che+17] to evaluate the LLM. The plaintext space of the CKKS scheme is $\mathbb{C}^n$, where $n$ is typically a power of two. Throughout this work, we use $n = 2^{16}$. The CKKS scheme supports the following basic operations that are used to construct all of the evaluation circuits.

- $\mathsf{ct}' \leftarrow \mathsf{EvalAdd}(\mathsf{ct}_1, \mathsf{ct}_2)$. This is the encrypted addition operation. If the input ciphertexts encrypt messages $m_1, m_2 \in \mathbb{C}^n$, then the output ciphertext $\mathsf{ct}'$ encrypts $m' \in \mathbb{C}^n$ where $m'[i] = m_1[i] + m_2[i]$.

- $\mathsf{ct}' \leftarrow \mathsf{EvalMult}(\mathsf{evk}, \mathsf{ct}_1, \mathsf{ct}_2)$. This is the encrypted addition operation. If the input ciphertexts encrypt messages $m_1, m_2 \in \mathbb{C}^n$, then the output ciphertext encrypts $m' \in \mathbb{C}^n$ such that $m'[i] = m_1[i] \cdot m_2[i]$.

- $\mathsf{ct}' \leftarrow \mathsf{EvalRotate}(\mathsf{evk}_\pi, \mathsf{ct}, \pi)$. This is the encrypted rotation operation where $\pi \colon [n] \to [n]$ is a permutation. If the input ciphertext encrypts the message $m \in \mathbb{C}^n$, then the output ciphertext $\mathsf{ct}'$ encrypts $m' \in \mathbb{C}^n$ such that $m'[i] = m[\pi(i)]$.

Observe that the $\mathsf{EvalMult}$ and $\mathsf{EvalRotate}$ both require evaluation keys. In addition, the evaluation key for $\mathsf{EvalRotate}$ is constructed with knowledge of the permutation $\pi$, and a different key is required for a different rotation. For algorithms with many different rotations, such as the bootstrapping operation described below, the size of these evaluation keys can become significant.

**Compute Levels & Bootstrapping.** A fundamental concept to understand in FHE performance is the notion of a compute level. An FHE ciphertext supports a finite number of compute levels before it must be refreshed to continue the computation. A ciphertext's compute levels are consumed primarily in the $\mathsf{EvalMult}$ operation, where each $\mathsf{EvalMult}$ consumes one level. If the two inputs to $\mathsf{EvalMult}$ have levels $\ell_1$ and $\ell_2$, then the output ciphertext will have level $\ell' = \min(\ell_1, \ell_2) - 1$. Once a ciphertext's levels have been consumed, further computation would result in a decryption failure. Instead, the ciphertext's levels must be refreshed in an operation called *bootstrapping* [Gen09]. Bootstrapping is an expensive operation that has been heavily studied [Bos+21; Cas+21; Jun+21].

The high-level paradigm for designing an FHE evaluation circuit is to first represent the desired function as an arithmetic circuit over $\mathbb{C}^n$, comprising of only Add, Mult, and Rotate gates. This circuit is then mapped to the encrypted domain, where each gate is replaced by its Eval counterpart. Finally, bootstrapping operations are placed in the circuit to ensure that no $\mathsf{EvalMult}$ operation is performed on a ciphertext that has no remaining compute levels.

**Polynomial approximations.** Natively, FHE only supports additions and multiplications on encrypted data. Other operations such as exponentiations, inverses, square roots and others, all needed for LLM evaluation, must be approximated using polynomial methods. We present in detail the approximations we make use of in Appendix D.

# D    Approximate Activation Functions

Since the CKKS scheme is designed to handle arithmetic operations, polynomial evaluation is easily supported. In contrast, functions like $\exp(\cdot)$ or $\tanh(\cdot)$ cannot be supported in a straightforward way. Following prior work on integer-only evaluation of deep learning models [Don+23; Zha+24], we approximate all functions required in the LLM evaluation with low-degree polynomials. Keeping the degree low is important as this minimizes the levels consumed in the polynomial evaluation, resulting in fewer bootstrapping calls. However, if the degree is too low then the approximation may not provide good accuracy. Below we discuss the approximations of different functions we use all throughout our work. These approximations are typically parameterized by different values that determine the degree and hence the respective accuracy. We discuss in Section 3 how we instantiate these parameters concretely.

Below we point out the *depth* of the resulting computation, which is what dictates the bottleneck when instantiated with FHE. Note that a degree-$D$ polynomial can be evaluated with depth $\log_2(D)$.

## D.1    Approximation of Comparison

We approximate the output of the sign function

$$\mathsf{sign}(x) = \begin{cases} -1 & x < 0 \\ 0 & x = 0 \\ 1 & x > 0 \end{cases}.$$

Arbitrary comparisons between $x$ and $y$ can be constructed by computing $\mathsf{sign}(x - y)$.

We use the techniques from [CKK20]. There, the approximation is given by $h(x) = f_n^{(d_f)} \circ g_n^{(d_g)}(x)$, where $f_n(x)$ and $g_m(x)$ are carefully chosen polynomials of degree $2n + 1$ and $2m + 1$ respectively. In [CKK20], $f_n(x)$ is given by

$$f_n(x) = \sum_{i=0}^{n} \frac{1}{4^i} \cdot \binom{2i}{i} x(1 - x^2)^i.$$

$g_m(x)$ on the other hand is not given in closed form. An algorithm for finding a suitable $g_m$ is given in [CKK20, Section 3.5], together with explicit examples for degree $3, 5, 7$ and $9$. These are the following:

$$\begin{cases} g_1(x) = -\frac{1359}{2^{10}} \cdot x^3 + \frac{2126}{2^{10}} \cdot x \\ g_2(x) = \frac{3796}{2^{10}} \cdot x^5 - \frac{6108}{2^{10}} \cdot x^3 + \frac{3334}{2^{10}} \cdot x \\ g_3(x) = -\frac{12860}{2^{10}} \cdot x^7 + \frac{25614}{2^{10}} \cdot x^5 - \frac{16577}{2^{10}} \cdot x^3 + \frac{4589}{2^{10}} \cdot x \\ g_4(x) = \frac{46623}{2^{10}} \cdot x^9 - \frac{113492}{2^{10}} \cdot x^7 + \frac{97015}{2^{10}} \cdot x^5 - \frac{34974}{2^{10}} \cdot x^3 + \frac{5850}{2^{10}} \cdot x. \end{cases}$$

Note that the composition requires depth $d_f \log(2n + 1) + d_g \log(2m + 1)$. We will make use of the $f$ and $g$ polynomials with degree $9$ (so $n = m = 4$), and we will typically set $d_f = d_g = 2$. Each of the $f$ and $g$ polynomials can be evaluated in

## D.2    Approximation of GeLU

We use the GeLU function [HG16] defined as

$$\mathsf{GeLU}(x) = 0.5x \left(1 + \tanh\left(\sqrt{2/\pi}\left(x + 0.044715x^3\right)\right)\right).$$

As in [Zha+24], we make use of the GeLU approximation from [Don+23], which consists of the following:

$$\mathsf{GeLU}(x) = \begin{cases} 0, & x < -4 \\ F_0(x), & -4 \leq x < -1.95 \\ F_1(x), & -1.95 \leq x \leq 3 \\ x, & x > 3 \end{cases} \tag{1}$$

This can be alternatively written as

$$(x \overset{?}{<} -4) \cdot (-F_0(x)) + (x \overset{?}{<} -1.95) \cdot (F_0(x) - F_1(x)) + (x \overset{?}{\leq} 3) \cdot F_1(x) + (3 \overset{?}{<} x) \cdot x$$

where we use the $\mathsf{sign}(x)$ approximation from above to perform the comparison. Here, the functions $F_0, F_1$ are:

$$\begin{aligned} F_0(x) = & -0.011034134030615728 \cdot x^3 - 0.11807612951181953 \cdot x^2 \\ & -0.42226581151983866 \cdot x - 0.5054031199708174 \\ F_1(x) = & \, 0.0018067462606141187 \cdot x^6 - 0.037688200365904236 \cdot x^4 \\ & +0.3603292692789629 \cdot x^2 + 0.5 \cdot x + 0.008526321541038084. \end{aligned}$$

From this, we see that the depth of the GeLU approximation is the depth required for approximating the comparison (since this is larger than the depth of $F_0$ or $F_1$), plus 1.

## D.3   Approximation of Layer Normalization

Recall that the $\mathsf{LayerNorm}$ operation, for $\boldsymbol{x} \in \mathbb{R}^d$, is defined as

$$\mathsf{LayerNorm}(\boldsymbol{y}) := \gamma \cdot \frac{\boldsymbol{x} - \mu}{\sqrt{\sigma^2 + \epsilon}} + \beta.$$

Here $\gamma, \beta, \epsilon \in \mathbb{R}$ are constants, $\mu = \frac{1}{d} \cdot \sum_{j=1}^{d} \boldsymbol{x}[j]$ and $\sigma^2 = \frac{1}{d} \cdot \sum_{j=1}^{d} (\boldsymbol{x}[j] - \mu)^2$. The value $\epsilon$ is a fixed small constant to avoid division by zero. where $\mu = \frac{1}{n} \sum_{i=0}^{n-1} a_i$ and $\sigma = \sqrt{\frac{1}{n} \sum_{i=0}^{n-1} (a_i - \mu)^2 + \epsilon}$, where $\gamma$ and $\beta$ are learned parameters and $\epsilon$ is a small constant. The core non-polynomial operation is given by $z \mapsto 1/\sqrt{z}$, for which we can use the inverse square root uses the techniques from [QX23]. We discuss these velow.

**Division by Square Root.**   The authors make use of Newton's iterative method. Once a starting approximation $y_0$ of $1/\sqrt{z}$ is chosen, iterate the following for $i = 1, \ldots, n$:

$$y_i = \frac{y_{i-1}(3 - z y_{i-1}^2)}{2},$$

with the final approximation being $y_n$. This has depth $3n$.

For choosing the initial point $y_0$, the authors first run a less accurate yet more efficient method. For this they propose two options: Taylor expansion, which is suitable for $x > 1$, and using Remez rational approximation, which is better for values that are close to 0. The work of [Zim+23] has found empirically that the variance (which is essentially the input to the square root) is large, so we use the Taylor expansion for the initial value.

For an approximation in the interval $[a, b]$, we choose an odd order Taylor expansion around $z_0 = (a + b)/2 + 1$ as the approximate initial value of $1/\sqrt{z}$. As suggested in [QX23, Section 5], we take degree 3 (which requires depth 2), so concretely this Taylor approximation looks like:

$$z \mapsto \frac{1}{\sqrt{z_0}} - \frac{z - z_0}{2\sqrt{z_0^3}} + \frac{3(z - z_0)^2}{8\sqrt{z_0^5}} - \frac{5(z - z_0)^3}{16\sqrt{z_0^7}}$$

## D.4   Approximation of SoftMax

For $\boldsymbol{x} \in \mathbb{R}^d$, $\mathsf{SoftMax}$ is defined as

$$\boldsymbol{y} = \frac{\exp(\boldsymbol{x}[i] - x_{\max})}{\sum_{j=0}^{m-1} \exp(\boldsymbol{x}[j] - x_{\max})},$$

where $x_{\max} = \max(\boldsymbol{x})$. The division by $e^{x_{\max}}$ is done in order to avoid large numerators and denominators.

**Exponentiation.** The approximation of exp is done via Taylor series, as in [Lu+23]:

$$\exp(x) \approx (1 + \frac{x}{2^r})^{2^r}, \quad x \leq 0,$$

where $r$, which corresponds to the resulting depth, is a parameter of choice.

**Max.** To compute $x_{\max}$, we first observe as in [CKK20] that

$$\max(a, b) = \frac{a + b}{2} + \frac{(a - b) \cdot \mathsf{sign}(a - b)}{2}.$$

For the sign function we can use the approximation from Section D.1. Once the max function of two values has been instantiated, obtaining the max of an array such as $\boldsymbol{x}$ can be done by using a binary tree of depth $\log_2(d)$, where $d$ is the dimension of $\boldsymbol{x}$. The resulting depth is $\log_2(d) \cdot (D + 1)$, where $D$ is the depth required by the sign approximation.

**Division.** Division uses Goldschmidt algorithm, which works as follows. To divide $A/B$, start with an approximation $F_0$ of $1/B$, and set $N_0 = A$ and $D_0 = B$. Then iterate $F_i \leftarrow 2 - D_{i-1}$, $N_i \leftarrow N_{i-1} \cdot F_i$ and $D_i \leftarrow D_{i-1} \cdot F_i$, for $i = 1, \ldots, d$. The output of the division is $N_d \approx A/B$. The depth of this approximation is $d$, since each iteration consumes one level.

In [ESF05], it is shown that, if $0 < F_0 < 2/B$, then the algorithm converges. We set $F_0 = 10$ as the initial estimate, which works well in our experiments.

### D.5 Handling Token Selection

As mentioned in Section B.3, one pass of the decoder architecture leads to a vector $\boldsymbol{b} \in \mathbb{R}^{\texttt{vocab\_size}}$, and the goal is to compute (homomorphically) the vector $\boldsymbol{e}_{i^*}$, where $\boldsymbol{e}_1, \ldots, \boldsymbol{e}_{\texttt{vocab\_size}}$ corresponds to the pre-trained embedding table, and $i^* \in [\texttt{vocab\_size}]$ is the ArgMax of $\boldsymbol{b}$, To this end, we use the approximation of ArgMax from [Zha+24], which works as follows. Let $\boldsymbol{u} \in \mathbb{R}^{\texttt{vocab\_size}}$ be the indicator vector which is $0$ in all entries, except for the index of the max $b_{\max}$ in $\boldsymbol{b}$, where it equals $1$. Then, we can compute $\boldsymbol{u}$ as

$$\boldsymbol{u} = (\mathsf{sign}(\boldsymbol{b} - b_{\max}) + 1)/2.$$

We can approximate $b_{\max}$ using the techniques from Section D.4, and the sign calls using the approach from Section D.1. The depth corresponds to the sum of the depths of these approximations. Finally, fetching $\boldsymbol{e}_{i^*}$ is done by taking the linear combination $\sum_{i=1}^{\texttt{vocab\_size}} \boldsymbol{u}[i] \cdot \boldsymbol{e}_i$.

# E   GPU Implementation of the CKKS FHE Scheme

In this section, we present our implementation of the CKKS FHE scheme. This implementation extends the popular and feature-rich OpenFHE library [AB+22] to use a GPU to accelerate the homomorphic operations. While we use this library to implement an LLM forward pass, this is the first open-sourced implementation of a GPU-accelerated CKKS scheme, which is of significant independent interest. The audience for this section is someone more familiar with the CKKS FHE scheme; this section can be safely skipped by those who are only interested in the LLM benchmarks. However, as a brief motivation for the focus on this function, the bootstrapping operation is at least $50\%$ of the runtime in all layers and typically closer to $80$-$90\%$ of the total time.

Our starting point for this implementation is the work of Jung et al. [Jun+21], which focuses on accelerating the bootstrapping implementation. The public portion of this code[5] is limited to the individual operations accelerated in their work, including the expensive number-theoretic transform (NTT) and RNS basis-change operations, rather than an end-to-end bootstrapping implementation. We incorporate these kernels into the OpenFHE CKKS bootstrapping code and implement further operations to connect these core functions and avoid any data movement off of the GPU. Our code includes an end-to-end bootstrapping implementation integrated into the OpenFHE API as well as all functions required to implement the LLM layers described above. This implementation inherits the improved accuracy from the careful tracking of the CKKS scaling factor in OpenFHE.

As prior works demonstrate [Cas+21], the bottleneck of CKKS bootstrapping quickly becomes the memory transfer if the compute accelerates faster than the local storage capacity. This is due to the size of the evaluation keys, which for bootstrapping can reach tens of gigabytes. For our benchmarks, we use a GPU with 80 GB of RAM, which allows us to cache all of the evaluation keys needed for bootstrapping and the subsequent LLM layers.

We present the benchmarks of our bootstrapping implementation in fig. 4. The CPU benchmarks were run on a machine with an Intel Xeon chip running at 2.4 GHz and 2 TB of RAM. The GPU benchmarks were run on the same machine and used an NVIDIA A100 80GB PCIeAll benchmarks were run within OpenFHE, which runs a depth 13 approximation of the CKKS modular reduction function. All bootstrapping hyperparameters were the same in all benchmarks. The level budget for the homomorphic encoding and decoding was set to 4 resulting in a total bootstrapping depth of 21. The number of decomposition digits was set to 3. The security level is at least 128 bits for the 10 and 20 output levels and 80 bits for 30 and 40 output levels. This is to accommodate the maximum modulus without growing the ring dimension.

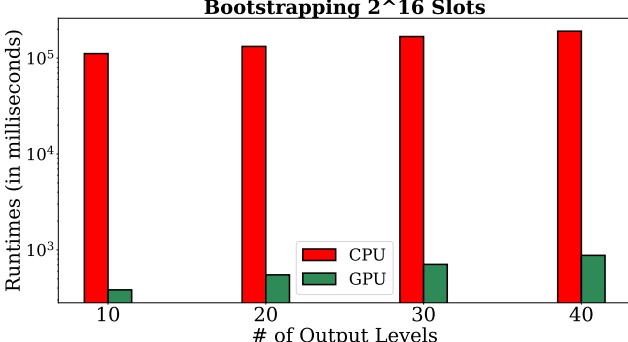

Figure 4: This figure presents a comparison between a CPU implementation of bootstrapping and a GPU implementation of bootstrapping for $n = 2^{16}$ slots with various output levels. Observe the log scale on the y-axis. The CPU implementation requires roughly 4-6 seconds per output level while the GPU implementation only requires 22-27 ms per output level, representing a speedup of $180$-$220\times$. These benchmarks are highly consistent with less than $5\%$ change over 10 iterations.

---

[5]https://github.com/scale-snu/ckks-gpu-core

# F   Future Work

While real-time chatbots under FHE remains out of reach, these benchmarks suggest that many applications are now practical to run in a secure way. This includes tasks that do not require real-time results, such as document summarizing or drafting (e.g. "Please write a speech for our CEO."). In addition, this performance improvement can translate to tasks that require the forward pass as a subroutine, such as fine-tuning a public model on private data. This training task is computationally expensive and often requires outsourcing, which can be safely enabled by this library. More concretely, a company may wish to train a more specialized LLM for a narrow task, such as an assistant for a technical role. The additional training data for this specialized task could easily be proprietary, and The resulting model can then be decrypted by the data owner or remain encrypted on the cloud for evaluation. These applications present numerous directions for future work.

## F.1   Simplifying the Activation Functions

These exciting applications motivate the study of models that are more optimized for the encrypted domain. In particular, the activation functions could likely be replaced with variants that are simpler to accurately approximate with a low-degree polynomial. A major candidate is the SoftMax function, which currently requires an expensive tree of comparisons to evaluate. However, this maximum value is only used to scale down an exponentiation input (i.e. to ensure that this input is negative). It seems very plausible that a trainable parameter could be included in the model to approximate an upper bound for this maximum value, removing the need for the comparison tree entirely. This would nearly eliminate the cost of the SoftMax function, requiring less than 1 bootstrapping operation.

The LayerNorm function could also potentially be replaced with the significantly cheaper BatchNorm function, which uses a pretrained mean and standard deviation for each window rather than computing these values on the fly. This would completely eliminate the cost of the LayerNorm function. Even if completely replacing this function is too ambitious, many approximation algorithms are significantly improved when provided a sufficiently accurate initial hint, and there is likely a middle-ground where a pretrained hint is used to reduce the complexity of LayerNorm approximation.

