# OpenReview forum: "Privacy-Preserving Large Language Model Inference via GPU-Accelerated Fully Homomorphic Encryption"
_NeurIPS.cc/2024/Workshop/SafeGenAi — SafeGenAi Poster_

### Official Review · Reviewer_3adq · 2024-10-09

**Rating:** 6
**Confidence:** 3

**Review:**

In this work, the authors propose a privacy-preserving inference of user requests using Fully Homomorphic Encryption (FHE) techniques.

Specifically, the user sends encrypted data to a third-party host containing the LLM, which processes the request and returns the output in encrypted form. The encrypted output is then sent back to the user, where the answer is decrypted.

However, several challenges exist, which the authors address in this work. Many activations in LLMs are not currently supported by FHE, to which the authors contribute an approximate implementation that is as close as possible, achieving a significant efficiency boost of 200x.

Moreover, the authors discuss contributing to the open-source library OpenFHE with their implementation on the GPT-2 model.

Overall, given the open-source contributions and the implementation of GPT-2 with polynomial approximation, the impact of this work is substantial. I would support for the acceptance.

---

### Official Review · Reviewer_39qa · 2024-10-09
**This study introduces an open-sourced implementation of GPU-accelerated FHE and proposal of  approximations of activation functions, maintaining minimal accuracy loss.**

**Rating:** 7
**Confidence:** 3

**Review:**

Strengths:
- The use of FHE for LLMs inference is a novel solution to address privacy concerns, particularly when sensitive data is involved. Implementing a GPU-accelerated version of FHE enhances its practicality and speed, achieving impressive performance improvement using GPU acceleratio. Moreover, the authors promise to provide open-source code for further use.
-  The paper presents a set of experiments using well-known datasets like LAMBADA, and ARC, offering insights into the trade-offs between accuracy and performance when employing polynomial approximations.

Limitations:
- Narrow Dataset Focus: The paper only benchmarks performance using three datasets (HellaSwag, LAMBADA, ARC), which are not necessarily representative of the wide variety of NLP tasks that LLMs are expected to handle. Any OOD dataset could provide a better evaluation of the model’s robustness, especially with the limitations mentioned.

There are minor grammatical errors, such as the use of _ran_ instead of _run_ and some sentences have awkward structures

Suggestion:  it would be great to include malicious actors and consider adversarial attacks

---

### Official Review · Reviewer_eHi6 · 2024-10-10

**Rating:** 4
**Confidence:** 3

**Review:**

**Summary**

This paper presents an implementation of fully homomorphic encryption (FHE) for transformer models. This is achieved by implementing existing methods / frameworks (CKKS, OpenFHE) on GPUs to enable faster LLM inference. The authors conduct experiments using GPT-2-small, showing that their FHE implementation produces negligible performance degradation on three datasets (Lambada, HellaSwag, and ARC) while being significantly faster than a CPU-only FHE baseline.

**Strengths**

- The motivation for this work is sound in that it aims to enable user privacy when interacting with an LLM hosted by a third party, which is increasingly becoming the norm as commercial models increase in popularity
- The results suggest that the implementation of FHE mechanically works in that it achieves significant speed improvements over the CPU baseline without significantly degrading model performance

**Weaknessess**

- Experiments are only conducted with GPT-2-small, a 117M parameter language model which is not close to SOTA LLMs in either performance or size. Even though the implementation works for GPT-2-small, there's no evidence that this will be consistent for larger models. I wonder if the approximations of activation functions necessary for FHE will result it greater performance degradation as the size of the model (and therefore the number of approximated activations) increases.
- Even with such a small model, the time for a forward pass is still a few minutes. Although the authors are correct to point out that this inference throughput could be acceptable for non-real-time applications, I question whether this would be practical for the more commonly used LLMs today which are orders of magnitude larger in size than GPT-2-small.
- The use of only three datasets for evaluating performance degradation makes the results not very convincing, especially when you consider that the evaluated model (GPT-2-small) performs quite poorly on these datasets.
- The paper is lacking a discussion or comparison of this approach to other related methods for preserving privacy in LLM applications, such as prompt randomization and secure multi-party computation.
- The main part of the paper is lacking technical details on modifications to the transformer that were implemented to enable FHE (e.g., approximations of activation functions). I suggest reducing the length of the introduction and moving these key technical details from the appendix to section 2.